# Taxonomically different symbiotic communities of sympatric Arctic sponge species show functional similarity with specialization at species level

Anastasiia Rusanova,[1,2,3] Viktor Mamontov,[1] Maxim Ri,[1] Dmitry Meleshko,[1,4] Anna Trofimova,[1,2] Victor Fedorchuk,[5] Margarita Ezhova,[6] Alexander Finoshin,[3] Yulia Lyupina,[3] Artem Isaev,[1] Dmitry Sutormin[1]

**ABSTRACT**  Marine sponges harbor diverse communities of associated organisms, including eukaryotes, viruses, and bacteria. Sponge-associated microbiomes contribute to the health of host organisms by defending them against invading bacteria and providing them with essential metabolites. Here, we describe the microbiomes of three sympatric species of cold-water marine sponges—*Halichondria panicea*, *Halichondria sitiens*, and *Isodictya palmata*—sampled at three time points over a period of 6 years in the White Sea. We identified the sponges as low microbial abundance species and detected stably associated bacteria that represent new taxa of sponge symbionts within Alpha- and Gammaproteobacteria. The sponges carried unique sets of unrelated species of symbiotic bacteria, illustrating the varying complexity of their microbiomes. At the community level, sponge-associated microbiomes shared common symbiotic features: they encoded multiple eukaryotic-like proteins, biosynthetic pathways and transporters of amino acids and vitamins essential for sponges. At the species level, however, different classes of eukaryotic-like proteins and pathways were distributed between dominant and minor symbionts, indicating specialization within microbiomes. Particularly, the taurine and sulfoacetate import and degradation pathways were associated exclusively with dominant symbionts in all three sponge species, suggesting that these pathways may represent symbiotic features. Our study indicates convergent evolution in the microbiomes of sympatric cold-water sponge species, as reflected by strong functional similarity despite the presence of distinct, taxonomically unrelated symbiotic communities.

Address correspondence to Dmitry Sutormin, d.a.sutormin@gmail.com.

The authors declare no conflict of interest.

See the funding table on p. 22.

*[This article was published on 16 October 2025 with an incorrect description of N. A. Pertsov. The error was corrected in the current version, posted on 28 October 2025.]*

**IMPORTANCE** Sponges are regarded among the earliest multicellular organisms and the most ancient examples of animal-bacterial symbiosis. The study of host-microbe interactions in sponges has advanced rapidly due to the application of next-generation sequencing (NGS) technologies that help overcome the challenges of investigating their communities. However, many sponge species, particularly those from polar ecosystems, remain poorly characterized. Here, we demonstrate that three sympatric cold-water sponge species, including two analyzed for the first time, harbor distinct sets of bacterial symbionts, stably associated over 6 years. Using CORe contigs ITerative Expansion and Scaffolding, an algorithm developed in this study, we reconstructed high-quality symbiont genomes and revealed shared features indicative of convergent evolution toward symbiosis. Notably, we identified a potentially novel symbiotic feature—a gene cluster likely involved in sulfoacetate uptake and dissimilation. We also observed shifts in microbiome composition, associated with increasing water temperatures, raising concerns about the impact of global warming on cold-water ecosystems.

**KEYWORDS**   sponges, holobiont, symbiosis, bacteria symbionts, 16S RNA, metagenomes, symbiotic features, global warming

Sponges, phylum Porifera, exhibit extensive global distributions across a wide range of habitats and display remarkable biodiversity (1). Sponges host complex communities of eukaryotes, prokaryotes, and viruses comprising a holobiont, wherein each participant contributes to the stability and functionality of the entire system (2, 3). Based on diversity and abundance of sponge-associated microbiomes (SAMs), sponge species are categorized as either high or low microbial abundance (HMA or LMA, respectively). HMA sponges can host more than a dozen distinct bacterial phyla with a total density reaching $10^8$–$10^{10}$ bacteria per gram of sponge wet weight, whereas less complex microbiomes of just two to five phyla with a total density of $10^5$–$10^6$ are typical for LMA sponges (4–6). To date, 40–60 different bacterial phyla have been identified in various sponge species (7, 8). Sponge-associated bacteria (SABs) are generally divided into transiently associated and stably associated groups. The latter are typically enriched in sponges compared with the surrounding water and exhibit high host specificity. Stably associated SABs are usually referred to as sponge symbionts, although functional characterization of host-bacterial interactions remains limited (9–11). This specificity makes the cultivation of sponge symbionts particularly challenging, as these bacteria are adapted to life within sponge tissues or cells. Consequently, metagenomics and metatranscriptomics serve as key tools for studying SAMs and predicting their functional traits (11).

Metagenomic studies have shown that most bacterial symbionts from different sponge species and various geographic locations share specific genomic traits (usually called symbiotic features) that reflect adaptation to a host and contribute to the stability of a holobiont. Among these features are the biosynthetic pathways and transporters of vitamins and amino acids, which are encoded in the genomes of sponge symbionts. It is suggested that bacteria provide these compounds to their hosts (3, 12, 13). At the same time, symbionts likely obtain certain nutrients from a host. For instance, the symbiotic bacterium *Candidatus* (*Ca.*) Taurinisymbion ianthellae is likely capable of utilizing taurine, potentially produced by a sponge, as a source of carbon, sulfur, and energy (14). Another example of symbiotic features includes genes encoding eukaryotic-like proteins (ELPs), which are enriched in SAM metagenomes and the genomes of symbiotic bacteria. ELPs are believed to mediate interactions with a host and are responsible for the recognition of symbionts and their discrimination from food-source bacteria (15–17).

Historically, most research on sponge microbiomes has focused on species from tropical and temperate regions, whereas polar sponges have remained relatively understudied, with a growing interest emerging only in recent years (18–23). Particularly, studies highlight the vulnerability of cold-water sponges and their microbiomes to global warming, which is an increasing concern (24–27). To study the long-term stability of polar SAMs and their functional traits, we applied metagenomics to three common sympatric sponge species (*Halichondria panicea* [HP], *Halichondria sitiens* [HS], and *Isodictya palmata* [IP]) from the White Sea over a sampling period of 6 years (in 2016, 2018, and 2022). Using 16S rRNA gene amplicon sequencing, we identified stably associated bacterial operational taxonomic units (OTUs) enriched in sponge microbiomes. We found fluctuations in the composition of the SAMs, which were affected by anomalously increased water temperatures during the 2018 summer season and were recovered by 2022. Using the hybrid shotgun metagenomic approach and the state-of-the-art core contigs iterative expansion and scaffolding (CORITES) refining algorithm, we obtained high-quality metagenome-assembled genomes (MAGs) of the symbiotic bacteria. Phylogenetic reconstruction assigned the identified symbionts to six novel species of Proteobacteria: *Ca.* Yagmuria paniceus, *Ca.* Ahtobacter symbioticus, *Ca.* Vellamobacter salmiensis, *Ca.* Vienanmeria sitiensis, *Ca.* Sampovibrio pertsovi, and *Ca.* Eurynomebacter symbioticus. Using genomic analysis, we revealed converging metabolic specializations within communities for a symbiotic lifestyle. Finally,

with fluorescence *in situ* hybridization (FISH) microscopy, we verified the presence of identified symbiotic bacteria within sponge tissues and found the species-specific localization of bacterial cells.

## MATERIALS AND METHODS

### Sample collection

For metagenomics, fragments of visually healthy *Isodictya palmata* (Ellis and Solander, 1786), *Halichondria sitiens* (Schmidt, 1870), and *Halichondria panicea* (Pallas, 1766) marine sponges were collected by scuba divers at 5 m–7 m water depth at the N. Pertsov White Sea Biological Station (WSBS Moscow State University [MSU], 66.5527°N, 33.1033°E, Kandalaksha Bay of the White Sea, Russia) in August to September 2016, 2018, and 2022 (Table S1). The sponge species were held separately for 2 h at 5°C in 5 L of seawater sterilized by filtering through a 0.22 µm filter (Sartorius). The identification of sponge species was performed by zoologist Boris Osadchenko (Lomonosov MSU) and further confirmed by 18S rRNA gene region amplification and sequencing (for 2016 samples; primer sequences in Table S2) or 18S rRNA gene reconstruction from metagenome assemblies (for samples from WSBS) (Fig. S1A; Table S3). As a control, 3 L of surrounding seawater at the sampling site was collected in a sterile container and immediately processed in the WSBS laboratory. Specimens of *Halichondria panicea* were also collected from the sublittoral zone at three different sites around Dalnye Zelentsy (69.10177°N 36.06234°E), Barents Sea, Russia, in August 2022 (Table S1).

### Analysis of microbiome 16S rRNA gene amplicon and shotgun metagenomic sequencing data

To isolate microbiome DNA from sponge samples, a 1 cm³ sponge tissue fragment was homogenized in sterile marine water, followed by bacterial enrichment using differential centrifugation and DNA extraction with the Diatom kit. For the marine water microbiome, 3 L of seawater was filtered through a 0.2 µm Sterivex filter, and DNA was extracted from the filter membrane. Details of sample processing and sequencing are provided in the Supplemental methods. 16S rRNA gene amplicon sequencing data analysis was performed as described earlier (28). Briefly, trimmed forward reads were processed with the DADA2 pipeline v.3.6.2 (29), and resultant amplicon sequence variants were clustered with MMseqs2 v.10-6d92c (30) into OTUs. Taxonomy was assigned to OTUs using SILVA v.138.1 (31). Principal coordinates analysis (PCoA), alpha-diversity, and taxonomic analyses were performed with the phyloseq package v.1.30.0 (32). To identify low-abundant SABs, OTUs meeting the following criteria were selected: (i) observed in most of the sponge samples of a particular species; (ii) with a relative abundance exceeding 1% in sponge samples; and (iii) with a relative abundance at least 50 times higher in a sponge sample than in seawater. A list of generated and reanalyzed 16S rRNA gene amplicon and shotgun data sets is provided in Table S3.

The methodology for assembly and binning of shotgun metagenomes is described in detail in the Supplemental methods. Briefly, Illumina or MGI-generated reads were assembled with SPAdes v.3.15.4 (33). Nanopore reads were assembled with Flye v.2.8.1-b1676 (34); SPAdes was used for hybrid assembly.

The obtained assemblies were binned with MaxBin 2.0 v.2.2.7 (35), CONCOCT v.1.1.0 (36), MetaBAT 2 v.2.12.1 (37), and binny v.0.2 (38). To reconstruct SAB MAGs, bins with completeness >50%, contamination <15%, and either a single 16S rRNA gene sequence matching an SAB OTU or no detectable 16S rRNA gene sequences were selected. SAB bins generated by different binners and classified identically by GTDB-Tk v.2.4.0 (39) were collected for refinement using CORITES (see Supplemental methods for the algorithm details), which is available on GitHub (https://github.com/sutormin94/CORITES).

The resulting SAB MAGs and medium-to-high-quality metagenomic bins of non-sponge-associated bacteria, obtained using binny from seawater and SAM metagenomes (2016 and 2018 samples), were used for annotation and further analysis.

## Functional annotation and phylogenetic reconstruction of MAGs

Open reading frames (ORFs) in the obtained metagenomic bins and reconstructed MAGs were annotated with MetaGeneMark2 v.1.23 (40) and Prokka v.1.14.6 (41). Protein-coding sequences were annotated against the Kyoto Encyclopedia of Genes and Genomes (KEGG) database using BlastKOALA v.3.0 (42), anvi'o v.8 (43), and the eggNOG-mapper online resource (44). Amino acid biosynthetic pathways were predicted with BlastKOALA v.3.0 and GapMind (45). Biosynthetic gene clusters (BGCs) were predicted with anti-SMASH v.7.0 (46). Secretion systems were detected with TXSScan (Galaxy version 2.0 + galaxy3) (47), and ELPs were identified using InterProScan v.5.64-96 (48) with the Pfam database (49). Signal peptides were predicted with SignalP 6.0 (50). The CRISPR-Cas systems were annotated using CRISPRCasTyper v.1.8.0 (51). Genetic cluster similarity was analyzed and visualized with Clinker (52). For predicting and comparing protein structures, we used AlphaFold 3 (53) and Dali (54).

The full-length 16S rRNA gene sequences derived from SAB MAGs were searched using blastn against NCBI nt, NCBI 16S rRNA, and SILVA NR99. The sequences retrieved from these databases were combined and clustered with MMseqs2 v.10-6d92c to remove duplicated sequences. Representative sequences were aligned using MUSCLE in MEGA-X (55), and a maximum likelihood tree was constructed based on the multiple alignments with 100 bootstrap iterations. A bootstrap consensus tree was visualized in iTOL (56).

Refined SAB MAGs were classified using the GTDB-Tk v.2.4.0. A tree constructed based on the concatenated alignment of 120 bacterial phylogenetically informative marker proteins (Bac120) was obtained and visualized in iTOL.

## Annotation of sponge transcriptomes

ORFs were predicted in the published assembled transcriptomes for HP (labeled, respectively, Tr1 and Tr2) (57, 58) and *Isodictya* sp. (27) with GeneMarkS-T v.5.1 (59), and unique sequences longer than 100 amino acids were selected for further analysis. Metabolic pathways were reconstructed using BlastKOALA and KEGG Mapper (60). To detect the expression of bacterial genes of sponge-associated bacteria, genes from the OTU4 and OTU23 MAGs were blasted against the HP transcriptomes. Hits were selected if their identity and coverage exceeded the 95% threshold.

## FISH microscopy

FISH was performed according to a standard protocol (61) with Cy3-labeled SAB-specific and Cy5-labeled bacterial universal probes (62) for 16S rRNA gene (Table S2). Briefly (see the detailed protocol in the Supplemental methods), sponge tissue samples were fixed in formaldehyde, dehydrated, and stored at −20°C until further processing. The rehydrated fragments were hybridized with an equimolar mixture of universal and particular SAB-specific probes. DNA was stained with Hoechst 33342. The stained samples were mounted with ProLong Gold antifade (Invitrogen) and imaged in Airyscan mode (63) using a Zeiss LSM 800 laser scanning confocal microscope at the Center of N.K. Koltsov RAS. Images were processed in Zen Black (Carl Zeiss), followed by analysis in ImageJ Fiji (64). The specificity of the SAB-specific probes was confirmed by hybridization with tissues from non-cognate sponge species, which showed no detectable signal.

## RESULTS

### Sympatric sponges harbor species-specific sets of host-associated OTUs

To study the long-term stability of Arctic sponge microbiomes and assess whether ecologically similar sponge species harbor similar or distinct communities, we selected

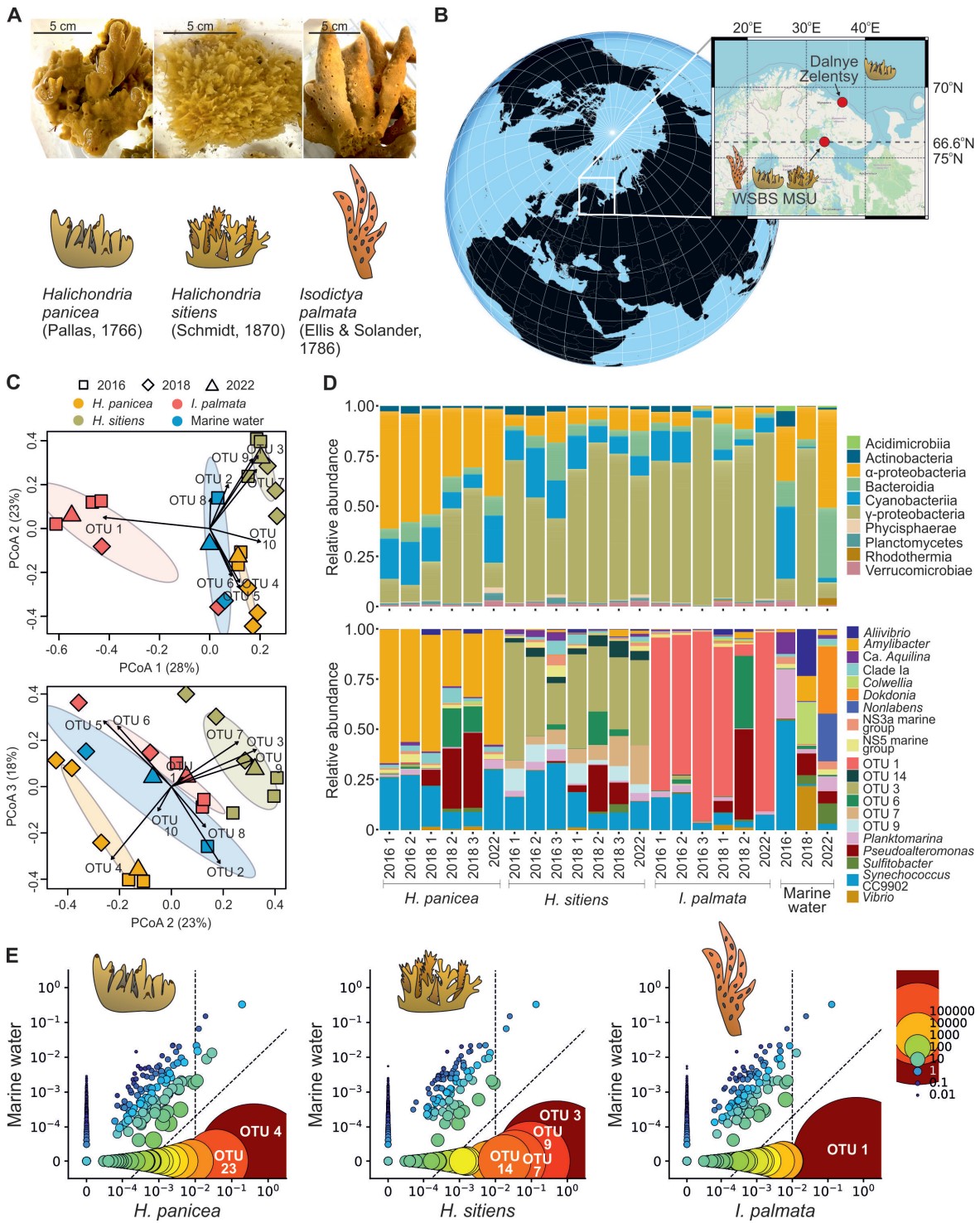

**FIG 1** Microbiomes of HP, HS, and IP. (A) Representative images of the three Porifera species studied. Scale bars of 5 cm are shown. Cartoon icons illustrating the sponge species are shown below the corresponding images. The icons are used throughout the manuscript as organism labels. (B) Map of the sample collection sites. (C) PCoA of Bray-Curtis dissimilarity between sponge and seawater microbiomes from the White Sea. (D) Relative abundance of top 10 bacterial classes (top) and top 20 genera (bottom) in sponge and seawater microbiomes from the White Sea. (E) Symbiont plots demonstrating the identification of SAB OTUs in the sponge microbiomes (data for representative replicates 2016_2 for HP; 2016_1 for HS; and 2016_1 for IP are shown). Each OTU represents a point. The radius and color of each point are proportional to the ratio of the relative abundances of the OTUs in sponge and seawater. The axes are presented on a logarithmic scale. The vertical dashed line marks a relative abundance of 1% in the sponge microbiome; the diagonal dashed line represents a 50:1 ratio between the relative abundances in the sponge and water microbiomes.

three abundant, sympatric cold-water marine sponges belonging to Demospongiae/Heteroscleromorpha—HP, HS, and IP (Fig. 1A). Samples were collected in Kandalaksha Bay of the White Sea in August to September of 2016, 2018, and 2022, along with seawater samples from the collection sites. Metagenomic DNA was extracted from enriched microbiome fractions and subjected to 16S rRNA gene V3-V4 region amplicon sequencing (Table S4). To investigate the geographical variability of HP microbiomes, we also collected sponges from Dalnye Zelentsy (the Barents Sea) (Fig. 1B). Rarefaction curves indicated that the samples were sequenced to saturation (Fig. S1B). Additionally, we reanalyzed publicly available metagenomic data sets for HP, *Isodictya kerguelenensis*, and *Isodictya erinacea* from various geographical locations in the Northern and Southern Hemispheres (Tables S5 to S7).

Beta-diversity analysis revealed that the studied sponges harbored distinct SAMs, which significantly differed in composition from the surrounding seawater (permutational multivariate analysis of variance [PERMANOVA] *P*-values <0.05, Fig. 1C; Fig. S2A and B). SAMs had comparable numbers of OTUs with seawater but lower Shannon and higher Simpson indices (Fig. S3), suggesting the presence of highly represented sponge-specific OTUs (Fig. 1D). This pattern indicates that the studied sponges belong to the LMA group (65, 66).

Indeed, the microbiome of HP from the White Sea was dominated by the genus *Amylibacter* OTU4 (Fig. 1D), sharing 100% identity with 16S rRNA gene from the HP symbiont *Ca.* Halichondribacter symbioticus (Table S8) (67). Similarly, it dominated the microbiomes of HP collected from geographically diverse locations, implying a tight symbiotic association with its host (Table 1) (68, 69). The HS microbiome was dominated by OTU3, which belongs to the UBA10353 order (Gammaproteobacteria) (Fig. 1D; Table 1; Table S8). OTU1 was dramatically overrepresented in the IP, except for one sponge collected in 2018 (Fig. 1D; Table 1). The OTU1 sequence perfectly matched one obtained from the sponge *Haliclona* sp. and had 98% and 96% identity with dominant OTUs from the microbiomes of the Antarctic sponges *I. kerguelenensis* (70) and *I. erinacea* (18), which may suggest symbiotic associations with sponges.

After determining the dominant species of the microbiome, we aimed to identify low-abundant SABs. Using the criteria for low-abundant SABs (see Materials and Methods), we found OTU23 (Puniceispirillales, Alphaproteobacteria), which was frequently associated with HP microbiomes from the White Sea and was absent in seawater (Fig. 1E; Tables S8 and S9). OTU23 was found in nearly half of the HP individuals collected from the Baltic Sea (14/26 samples) (69) and was also identified with Pebblescout (71) in shotgun metagenomes of HP collected in Scotland. Interestingly, it was scarce in samples from the North Atlantic (14/84 samples) and the Barents Sea (0/4 samples). Concordantly, OTU23 bacterium was significantly more abundant in samples from the White Sea and Baltic Sea compared to samples from the Barents Sea and North Atlantic (Fig. S2C), indicating that OTU23 is likely not an essential member of the HP microbiome but is frequently associated with this sponge in some geographical locations (Table 1). Similarly, we detected three additional OTUs associated with the HS sponges (Fig. 1E; Table 1; Table S9)—OTU7 (Gammaproteobacteria, Pseudomonadales), OTU9 (Gammaproteobacteria, HOC36), and OTU14 (Gammaproteobacteria), two of which, OTU9 and OTU14, shared a high level of identity with 16S rRNA gene sequences obtained from other sponge species (Table S8). No additional OTUs were found to be associated with IP (Fig. 1E).

Repeated sampling of sponge microbiomes from the WSBS location allowed us to observe their long-term dynamics. The identified SAB OTUs were stably associated with sponge species over the 6-year period (Fig. 1D; Fig. S2A). However, in the 2018 data, several SAB OTUs (OTU1, OTU3, OTU14, and OTU23) were also detected at low levels in seawater (<0.01% relative abundance), whereas OTU4 reached a relative abundance of 8%. In non-host sponges, these OTUs were observed at low levels (average relative abundance <0.15%) across all sampling time points (Fig. S2A). Despite bacterial exchange between sympatric sponges, no new prolific interactions were established,

**TABLE 1** Relative abundance of identified SAB OTUs in 16S rRNA amplicon sequencing data sets and their taxonomy according to the SILVA database

| Sponge species | OTU ID | Geographical location | Relative abundance ± SD, % | Number of samples | Taxonomy (SILVA 138.1) |
|---|---|---|---|---|---|
| *H. panicea* | OTU4 | White Sea | 33 ± 11 | 6 | Bacteria/Proteobacteria/Alphaproteobacteria/ |
| | | North Atlantic | 39 ± 16 | 84 | Rhodobacterales/Rhodobacteraceae/*Amylibacter* (g) |
| | | Baltic Sea | 48 ± 11 | 26 | |
| | | Barents Sea | 50 ± 4 | 4 | |
| | OTU23 | White Sea | 2 ± 1 | 6 | Bacteria/Proteobacteria/Alphaproteobacteria/ |
| | | North Atlantic | 0 ± 0[a] | 84 | Puniceispirillales/SAR116 clade (f) |
| | | Baltic Sea | 0.5 ± 0.7 | 26 | |
| | | Barents Sea | 0 ± 0 | 4 | |
| *H. sitiens* | OTU3 | White Sea | 30 ± 11 | 7 | Bacteria/Proteobacteria/Gammaproteobacteria/ UBA10353 marine group (o) |
| | OTU7 | White Sea | 7 ± 3 | 7 | Bacteria/Proteobacteria/Gammaproteobacteria/ Pseudomonadales/OM182 clade (f) |
| | OTU9 | White Sea | 5 ± 3 | 7 | Bacteria/Proteobacteria/Gammaproteobacteria/HOC36 (o) |
| | OTU14 | White Sea | 3 ± 2 | 7 | Bacteria/Proteobacteria/Gammaproteobacteria (c) |
| *I. palmata* | OTU1 | White Sea | 57 ± 28 | 6 | Bacteria/Proteobacteria/Gammaproteobacteria/ Burkholderiales/EC94 (f) |

[a]OTU23 was detected in some *H. panicea* samples from the North Atlantic with an average relative abundance of 0.004 ± 0.01%. (s), species; (g), genus; (f), family; (o), order; (c), class.

indicating a specific symbiosis. Worth noting, we observed an increased abundance of Alteromonadales and Vibrionales OTUs in sponge and seawater samples collected in 2018, which may indicate ecological perturbations during that season (Note S1; Fig. S2D).

Taken together, our 16S rRNA gene amplicon sequencing data suggest that the studied sympatric Arctic sponge species harbor distinct SAMs, dominated by strongly host-associated species-specific OTUs, which likely represent bacterial symbionts.

## Recovery of SAB MAGs and their refining with CORITES

To reconstruct SAB genomes, we sequenced sponge metagenomes with short- and long-read technologies and obtained initial metagenomic bins using several commonly used binners. Bins obtained from long-read assemblies were more continuous; however, they typically contained several different 16S and/or 23S rRNA gene sequences, indicating contamination. To avoid possible chimeric sequences, long-read assemblies were excluded from further analysis.

Multiple NGS datasets generated over the 6-year sampling period allowed us to obtain numerous bins for identified SABs. Phylogeny reconstruction using GTDB-Tk demonstrated clustering of related bins, indicating a robust classification independent of the dataset and/or binning algorithm used (Fig. S4). Due to the differences between binning approaches, related bins obtained with different binners from the same metagenome had a considerable fraction of unshared contigs (on average, 62% ± 28% of the contigs were shared). To reduce the frequency of binning errors, and to refine and scaffold bins, we developed and applied the CORITES algorithm (Note S2). Using CORITES, we were able to increase completeness and reduce contamination for most sponge-associated MAGs compared with the initial bins (Table S10). The reconstructed MAGs had high completeness (>80%, except for OTU23 from HP), low contamination (<5%), and nearly complete sets of tRNA genes, indicating their medium-to-high quality by the minimum information about a metagenome-assembled genome (MIMAG) criteria (72) (Table S11). A low completeness level (63%) and the small size of the OTU23 MAG can indicate incomplete assembly or genome reduction due to a potential symbiotic lifestyle.

Overall, we were able to reconstruct MAGs for all SABs identified with 16S rRNA gene amplicon sequencing.

## The identified SABs represent new bacterial taxa of sponge symbiotic bacteria

The phylogenetic analysis of the recovered SAB MAGs and their full-length 16S rRNA gene sequences allowed us to infer their taxonomy.

Both methods unambiguously assigned the HP-dominating OTU4 SAB to *Ca*. H. symbioticus (Fig. 2). The MAG of the minor HP-associated bacterium, OTU23, was classified as a new member of the genus JAGWAQ01 within the UBA1172 family (Puniceispirillales, Alphaproteobacteria) (Fig. 2). According to the 16S rRNA gene sequence, the OTU23 MAG belonged to a clade of sponge- and coral-associated bacteria known as "Calcibacteria" (73) and to a larger clade that is sister to the Puniceispirillaceae family (Fig. S5). We propose the new name *Ca*. Yagmuria as a replacement for the JAGWAQ01 genus and the species name *Ca*. Yagmuria paniceus for the OTU23 bacterium: the genus name commemorates our colleague Eldar Yagmurov, who passed in 2022, and the species name reflects the host name. For the UBA1172 family, we propose the replacement name *Ca*. Calcibacteraceae after the "Calcibacteria" group (Table 2).

The dominant HS-associated OTU3 MAG belonged to the order Arenicellales (Gammaproteobacteria) and represented the initial member of a novel family based on GTDB-derived taxonomy and a relatively low relative evolutionary divergence (RED) value (Table 2; Fig. 3). Phylogenetic reconstructions based on a full-length 16S rRNA gene sequence revealed two potential members of this family isolated from marine sediments (Fig. S6). We propose the family name *Ca*. Ahtobacteriaceae, with the species type *Ca*. Ahtobacter symbioticus for the OTU3 bacterium, named after the sea god Ahto from Karelo-Finnish folklore. The species epithet "symbioticus" is related to the lifestyle of OTU3 in the HS sponge.

The HS-associated OTU7 MAG was assigned to the genus UBA9659 (*Ca*. Azotimanducaceae/HTCC2089, Gammaproteobacteria), comprising primarily free-living bacteria from seawater (Fig. 3). Correspondingly, sequences related to the OTU7 full-length 16S rRNA gene were isolated from seawater and freshwater environments (Fig. S7). We propose the replacement name *Ca*. Vellamobacter for the UBA9659 genus and species name *Ca*. Vellamobacter salmiensis for the OTU7 bacteria. The genus name is derived from "Vellamo," a goddess of water in Karelo-Finnish folklore. The species name indicates the location (the Great Salma Strait of the White Sea) where the samples were collected (Table 2).

The HS-associated OTU9 MAG was classified as a representative of a new genus within the Porifericomitaceae family, order *Ca*. Porifericomitales/UBA6729 (Gammaproteobacteria), a group known for its association with sponges (75) (Fig. 3). Correspondingly, the full-length 16S rRNA gene sequence reconstructed for OTU9 belonged to *Ca*. Porifericomitales and was most closely related to a sequence isolated from *Hymeniacidon heliophila* sponge (95% identity) (Fig. S8). For OTU9, we propose a new genus name, *Ca*. Vienanmeria, and the species name *Ca*. Vienanmeria sitiensis. The genus name is derived from "Vienanmeri," Karelian and Finnish name for the White Sea, whereas the species name reflects the host in which it was identified (Table 2).

The HS-associated OTU14 MAG was assigned to a novel sponge-associated family *Ca*. Oxydemutatoceae (75) within Porisulfidales (76) (Gammaproteobacteria) (Fig. 3). Its full-length 16S rRNA gene sequence clustered with a 16S rRNA gene clone isolated from the sponge *Ectyoplasia ferox*, which is basally positioned within *Ca*. Oxydemutatoceae (Fig. S9). We propose the name *Ca*. Sampovibrio pertsovi for OTU14. The genus was named after the "Sampo," the magical object in the Karelo-Finnish epic Kalevala, believed to be a source of happiness and well-being. "Vibrio" reflects the morphology of this bacterium (see the FISH section, Fig. 7F). The species name honors Dr. N. A. Pertsov (1924–1987), a distinguished former director of WSBS MSU (Table 2).

The IP-associated OTU1 MAG was classified as a new member of the JANQAO01 genus belonging to the family of sponge- and coral-associated bacteria recently described as *Ca*. Persebacteraceae (*Ca*. Tethybacterales, Gammaproteobacteria) (74) (Fig. 3). Consistent results were obtained using 16S rRNA gene phylogeny (Fig. S10). We propose

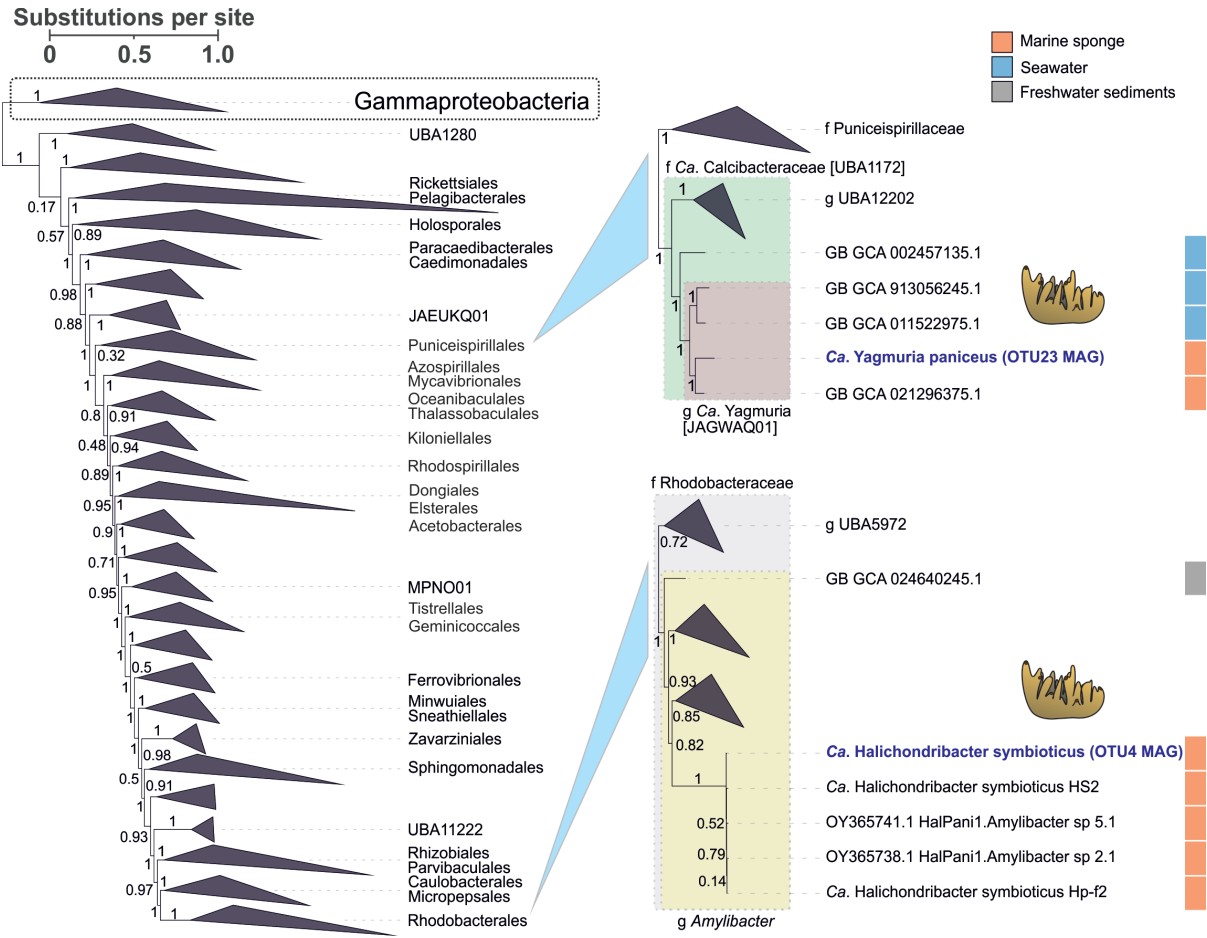

**FIG 2** Maximum-likelihood tree of Alphaproteobacteria. The Gammaproteobacteria clade is shown as an outgroup. Local support values obtained with the Shimodaira-Hasegawa test are indicated for nodes (1,000 resamples). The SAB MAGs obtained in this study are highlighted in blue. Host sponges are indicated with cartoon icons. The isolation sources of MAGs are color-coded.

a new genus name *Ca*. Eurynomebacter as a replacement name for the JANQAO01 genus and the name *Ca*. Eurynomebacter symbioticus for the OTU1 bacteria specifically, following the naming logic proposed by Taylor et al. for Tethybacterales (74). Eurynome is an Oceanid, and a species epithet "symbioticus" reflects the ecological niche of this bacterium (Table 2).

Four out of seven studied bacteria belonged to clades of sponge- or coral-associated bacteria, suggesting a symbiotic lifestyle and linked genomic adaptations. The other three bacteria belonged to mixed (*Ca*. H. symbioticus OTU4) or free-living (*Ca*. A. symbioticus OTU3 and *Ca*. V. salmiensis OTU7) groups of bacteria, presumably indicating evolutionarily recent (or facultative) symbiosis with sponges. For the sake of simplicity, we will further refer to the described species by the original OTU number.

## General metabolic characteristics of the reconstructed sponge-associated MAGs

To gain insights into the functional potential of SABs and detect possible metabolic interactions within a holobiont, we analyzed their metabolic capabilities. By reconstructing putative pathways in sponge-associated MAGs using KEGG, we identified nearly complete glycolysis (M00001) and citric acid cycle (TCA; M00009) pathways present in most of them. The OTU14 MAG (HS), however, lacked the pyruvate dehydrogenase complex (M00307), the TCA cycle, and its variation, the glyoxylate cycle (M00012), possibly indicating genomic reduction (Fig. 4A). An incomplete pentose phosphate

**TABLE 2** Classification of the identified bacterial sponge symbionts using GTDB and the proposed naming for novel taxa[c]

| Sponge | OTU | RED[d] value | GTDB taxonomy | Proposed taxon names | Clade ecology[b] |
|--------|-----|-------------|---------------|---------------------|------------------|
| *H. panicea* | OTU4 | 0.99 | o Rhodobacterales; f Rhodobacteraceae; g *Amylibacter/Halichondribacter*; s *symbioticus*[a] | No name assigned | Mixed |
| | OTU23 | 0.94 | o Puniceispirillales; f UBA1172; g JAGWAQ01; s | f *Ca.* Calcibacteraceae; g *Ca.* Yagmuria; s paniceus | Sponge- or coral-associated |
| *H. sitiens* | OTU3 | 0.74 | o Arenicellales; f; g; s | f *Ca.* Ahtobacteriaceae; g *Ca.* Ahtobacter; s symbioticus | Free-living |
| | OTU7 | 0.97 | o Pseudomonadales; f Azotimanducaceae; g UBA9659; s | g *Ca.* Vellamobacter; s salmiensis | Free-living |
| | OTU9 | 0.81 | o Porifericomitales; f Porifericomitaceae; g; s | g *Ca.* Vienanmeria; s sitiensis | Sponge- or coral-associated |
| | OTU14 | 0.86 | o Porisulfidales; f Oxydemutatoceae; g; s | g *Ca.* Sampovibrio; s pertsovi | Sponge- or coral-associated |
| *I. palmata* | OTU1 | 0.86 | o AqS2/Tethybacterales; f AqS2/Persebacteraceae; g JANQAO01; s[a] | g *Ca.* Eurynomebacter; s symbioticus | Sponge- or coral-associated |

[a]The GTDB taxonomy was extended with taxa proposed in references 67 and 74.
[b]A bacterial clade was assigned to sponge- or coral-associated if related 16S gene sequences were predominantly isolated from these animals, to free-living if sequences were predominantly obtained from marine sediments or water, to mixed if a fraction of sequences was obtained from sponges or corals.
[c]s, species; g, genus; f, family; o, order.
[d]RED, relative evolutionary divergence.

pathway (M00004) was detected in several MAGs from different sponges. Autotrophic carbon fixation was not reliably identified in any of the MAGs. Although the completeness of the Calvin cycle (M00165) and the reductive TCA cycle (M00173) in some genomes, such as *Ca*. H. symbioticus from the HP (see (Note S3), reached 70%, the same genes are also involved in oxidative pathways. Therefore, we inferred that all identified SABs are heterotrophic. SAB MAGs also carried genes encoding cytochrome *c* oxidase (M00155), other components of the respiratory electron transport chain, and ATP synthase (M00157), which are likely utilized for ATP production (Fig. 4A). No nitrogen metabolism pathways (such as nitrogen fixation, nitrification, denitrification, nitrate reduction, or anammox) were detected in the MAGs (77).

We detected diverse sulfur metabolism pathways enriched in SAB MAGs from HS. The OTU3 MAG encoded the *dsr* operon (*dsrABLEFHCMKJOP* genes), which is likely responsible for the oxidation of sulfide to sulfite via the reverse dissimilatory sulfate reduction (DSR) pathway, as it lacked the *dsrD* gene and included the *dsrEFH* genes (78). Additionally, this MAG contained genes of the sulfur oxidation (SOX) system (79)—the *soxABXYZ* operon responsible for the oxidation of thiosulfate to sulfate—and lacked *soxCD* genes (Fig. 4A and B). The presence of both the DSR and SOX pathways suggests that the OTU3 bacterium may utilize sulfide and thiosulfate as electron donors, as was hypothesized for two unrelated sponge symbionts with a similar composition of pathways (76, 80). The genome of the OTU14 MAG encoded a complete reverse-DSR sulfur oxidation pathway and carried *soeABC* genes for the oxidation of sulfite to sulfate (81), flavocytochrome C dehydrogenase (*fccAB* genes) for sulfide oxidation to elemental sulfur (82), and a partial *sox* operon (*soxXY* genes) (Fig. 4A and B). The OTU9 MAG also contained a partial *sox* operon (*soxCDY*), which may be attributed to incomplete genome assembly. In contrast to other HS SABs, the OTU7 MAG lacked pathways of sulfur oxidation but had a complete assimilatory sulfate reduction (ASR) pathway, reducing sulfate to sulfite and further to sulfide (83). Among bacteria from other sponges, only the OTU23 MAG (HP) had genes related to sulfur metabolism and contained the *soeABC* operon (Fig. 4A and B). The enrichment of sulfur transformation pathways, particularly oxidation, in HS SABs suggests the specialization of this community in sulfur metabolism and its potential role in sulfur cycling within the holobiont.

Next, we estimated completeness of all KEGG metabolic modules (and taurine and sulfoacetate metabolic pathways added independently) in the identified SAB MAGs and in the most closely related genomes (listed in Table S13), using anvi'o, and compared their predicted metabolic capabilities (Fig. S12A). We observed that OTU23, OTU9,

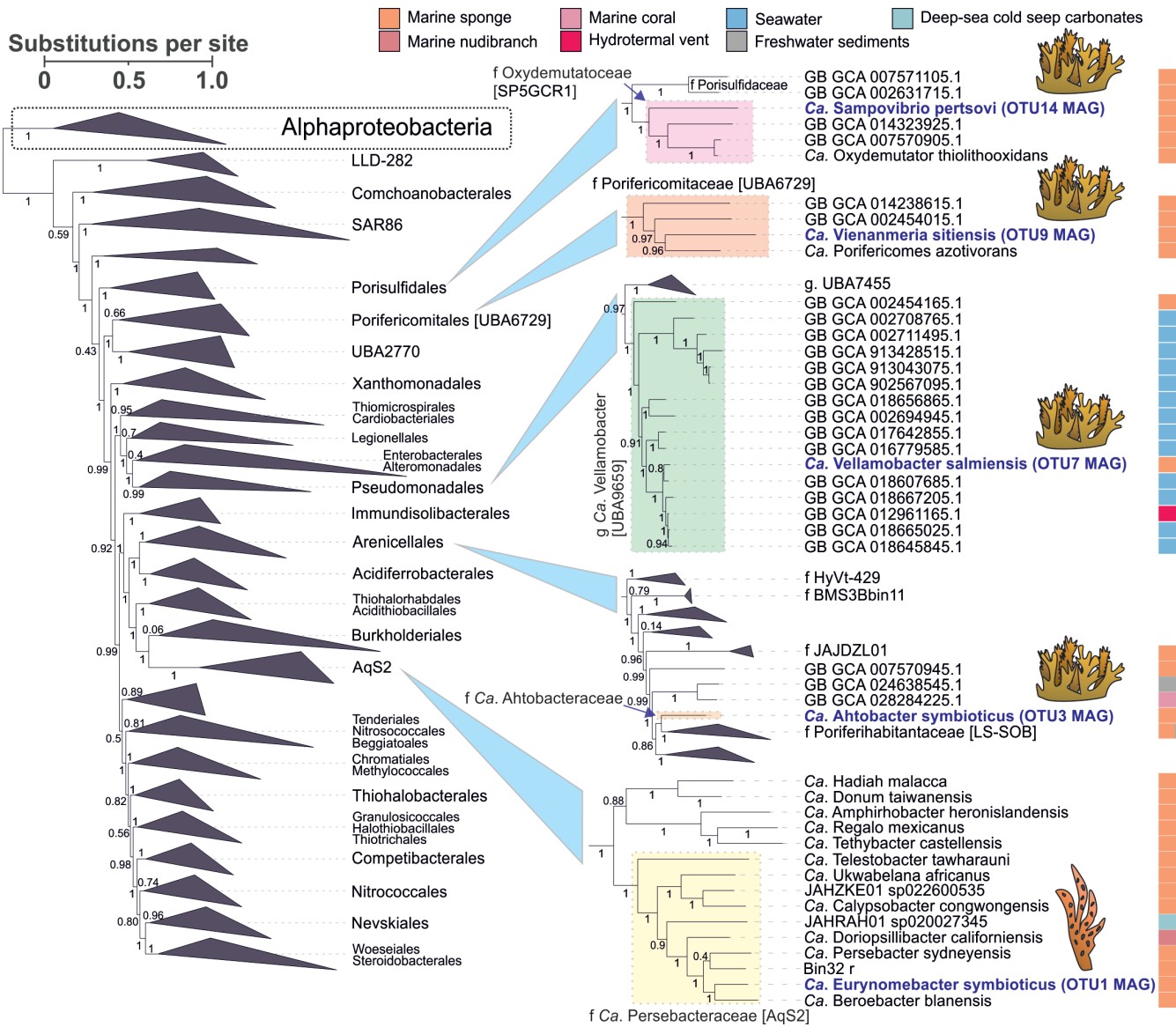

**FIG 3** Maximum-likelihood tree of Gammaproteobacteria. The Alphaproteobacteria clade is shown as an outgroup. Local support values obtained with the Shimodaira-Hasegawa test are indicated for nodes (1,000 resamples). The sponge-associated MAGs obtained in this study are highlighted in blue. Host sponges are indicated with cartoon icons. The isolation sources of the MAGs are color-coded.

and OTU14 genomes exhibited distinct patterns of module completeness compared to related genomes, as revealed with the principal component analysis (PCA) (Fig. 4C; Fig. S12B). These SAB MAGs had decreased completeness of both genomes and metabolic modules compared to related genomes, which underlies the observed differences (Fig. S12C). This decreased completeness may reflect genome reduction, but it also may be an artifact of MAG reconstruction. The latter scenario seems more plausible for OTU9 and OTU14, as both belong to well-known symbiotic groups—Porifericomitales and Porisulfidales, respectively. Other SAB MAGs had comparable completeness and similar composition of metabolic modules to their closest known relatives (Fig. 4C; Fig. S12B). Nevertheless, we identified several pathways that were nearly complete in SAB MAGs but less complete in related genomes. As such, the OTU1 MAG had a fully complete Entner-Doudoroff pathway (M00008) and TCA cycle (M00009), unlike other Tethybacterales. The OTU9 MAG possessed a nearly complete pimeloyl-ACP biosynthesis pathway (M00572) and a complete glycine cleavage system (M00621), compared

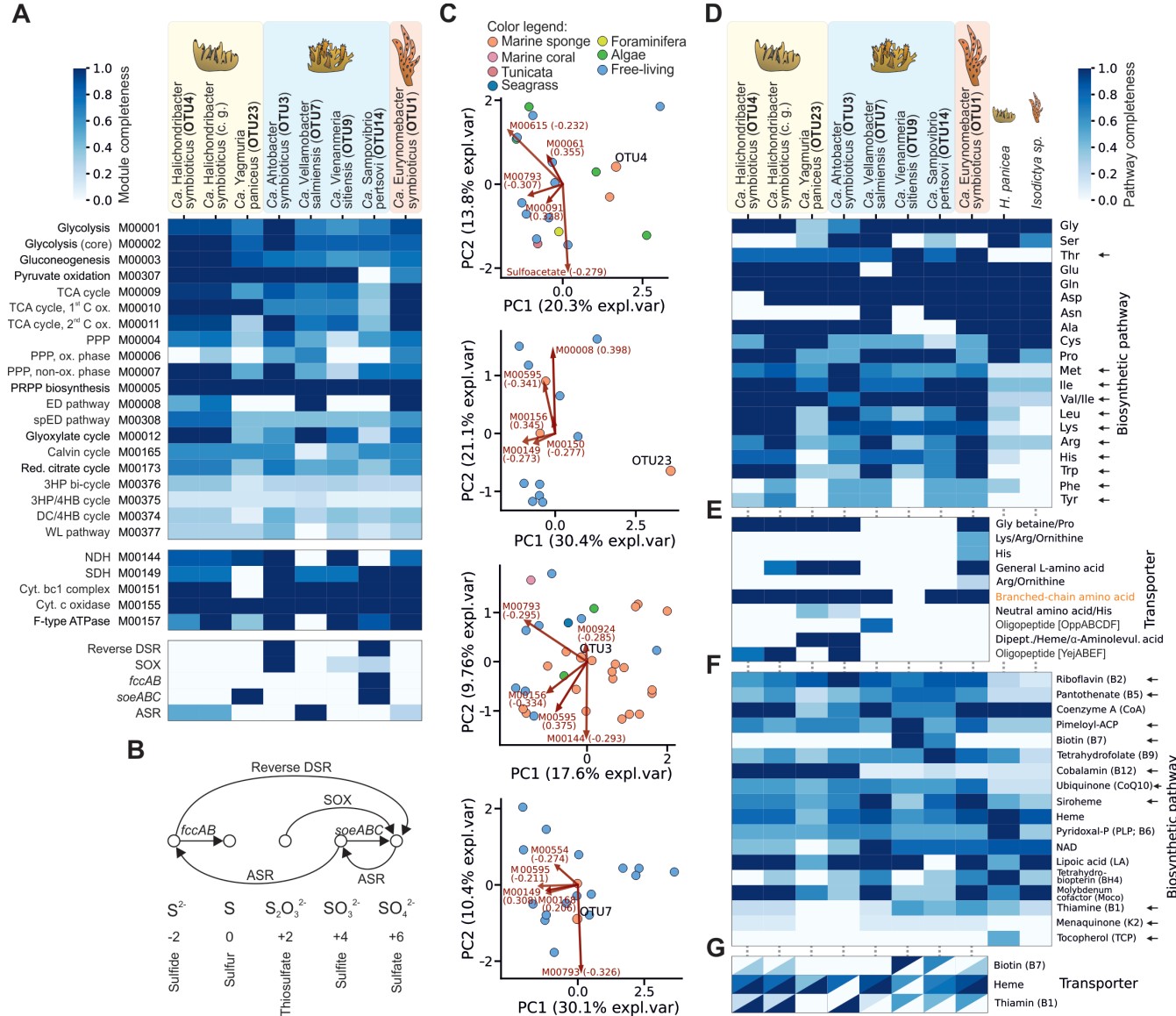

**FIG 4** Predicted metabolic capabilities of identified SAB MAGs. (A) Heatmap showing the completeness of general metabolic pathways. The OY365741.1 genome, identified as *Ca.* H. symbioticus, was included in the analysis. (B) A scheme of common sulfur metabolism pathways. ASR, assimilatory sulfate reduction; DSR, dissimilatory sulfate reduction; SOX, sulfur oxidation system; c.g., complete genome. (C) Principal component analysis biplots of KEGG module completeness in SAB MAGs (labeled dots) and taxonomically related bacteria. Isolation source of bacteria is color-coded. Data are shown for OTU4, OTU23, OTU3, and OTU7. (D) Heatmap showing the completeness of *de novo* amino acid biosynthetic pathways identified in SABs and sponge transcriptomes. (E) Completeness of gene sets related to amino acid transport. (F) Completeness of *de novo* vitamin/cofactor biosynthetic pathways. (G) Completeness of gene sets related to vitamin/cofactor transport. In all heatmaps, color indicates the completeness of gene sets, ranging from 0 (no genes of a pathway/transporter detected) to 1 (a complete pathway/transporter observed). Ordering of genomes is identical in panels D–G. Arrows in panels D and F indicate pathways not detected in sponge transcriptomes. In panel G, the heatmap values in the upper and lower triangles represent the completeness of the biosynthetic pathways for vitamins and their corresponding transporters, respectively.

to other Porifericomitaceae. The OTU23 MAG had a nearly complete leucine degradation (M00036) and formaldehyde assimilation (M00344) pathways, in contrast to other UBA1172 genomes. The OTU7 MAG encoded a nearly complete urea cycle (M00029), compared to other UBA9659 genomes. The OTU3, OTU4, and OTU14 MAGs did not have any distinct pathways when compared to other Arenicellales, *Amylibacter*, and Porisulfidales genomes, respectively. Among the SABs that had closely related free-living counterparts (OTU4/OTU23/OTU3/OTU7), only OTU3 MAG had several nearly complete

pathways that were less complete in its free-living relatives: homoprotocatechuate degradation (M00533), D-galactonate degradation (M00552), and cobalamin biosynthesis (M00924) pathways.

We conclude that most SAB MAGs exhibited metabolic capabilities similar to closely related genomes, except OTU23 (a potential genome reduction) and OTU3 (an acquisition of several pathways), which may represent a recent adaptation to symbiotic lifestyle.

## SABs may complement host metabolism

To identify metabolic capabilities that are absent in host sponges and potentially complemented by SABs, we analyzed publicly available transcriptomes of HP (Tr2) and *Isodictya* sp. (Note S4). No transcriptomic data have been published for HS, but considering its close relationship to HP (99.6% identity by 18S), we assumed that they may share similar metabolic potential.

We analyzed the biosynthetic pathways of proteinogenic amino acids in the HP and *Isodictya* sp. transcriptomes and found no expression of pathways for branched-chain (Ile, Leu, Val), non-polar (Met), polar (Thr), positively charged (Arg, Lys), and aromatic (His, Trp, Phe, Tyr) amino acid pathways, suggesting that these amino acids are essential for sponges. At the same time, these nearly complete pathways were identified in the MAG of at least one SAB from each of the three studied sponge species, except for the Phe/Tyr biosynthetic pathway (Fig. 4D). Further prediction of Phe/Tyr biosynthetic pathways in SAB MAGs using GapMind revealed nearly complete pathways in HS SABs (OTU3, OTU7, and OTU14) and in the IP SAB OTU1. Moreover, this pathway was also found in the *Ca*. H. symbioticus genomes (OY365741.1 and OY365738.1), suggesting that its absence in the OTU4 MAG from HP was due to genome incompleteness.

Next, we analyzed the presence of genes encoding amino acid and peptide transporters in SAB MAGs. SAB MAGs from all three microbiomes harbored genes encoding a general L-amino acid ABC transporter, and all SAB MAGs, except OTU9 (HS), encoded an ABC transporter for branched-chain amino acids (Fig. 4E). Moreover, we reanalyzed the published HP Tr1 (meta)transcriptome (Note S4) and detected the expression of branched-chain amino acid and general L-amino acid transporters for both OTU4 and OTU23 SABs. Since ABC transporters can function bidirectionally (84, 85), SABs may potentially supplement their hosts with essential amino acids (particularly branched-chain ones). Taken together, our analysis suggests a potential complementation of sponge requirements for essential amino acids by SABs.

Interestingly, a bacterial consortium in HS showed a division of biosynthetic functions: complete Met and Leu pathways were found exclusively in OTU3 MAG, while Arg and Trp pathways were present only in the OTU7 MAG. The OTU9 MAG was the only one lacking the Phe and Tyr pathways, while nearly complete pathways for other amino acids (Ile, Val, Lys, His, and Thr) were present in all four MAGs. This specialization may explain the relatively complex community in HS and suggests potential cross-feeding among its SABs. In contrast, the OTU1 MAG, the sole SAB from the IP, has nearly complete or complete pathways for all the above-mentioned presumably essential amino acids, indicating that it can likely fulfill the metabolic needs of its host alone.

Similarly, we analyzed the potential complementation of biosynthetic pathways of vitamins and cofactors between MAGs and sponge transcriptomes. The biosynthetic pathways of several B vitamins (B1, B2, B5, B7, and B12), pimeloyl-ACP, ubiquinone, and siroheme were scarce in both sponge transcriptomes. Among them, nearly complete or more complete, if compared to sponge, pathways for B2, B5, pimeloyl-ACP, ubiquinone, and siroheme were found in a MAG of at least one SAB from each of the three studied sponge species (Fig. 4F). Pathways for B1 were identified in OTU1 MAG (IP) and OTU9 (HS), but not in MAGs from HP, suggesting an alternative source of this vitamin for this sponge species. Similarly, B12 pathways were only present in SAB MAGs from HP and HS, but not in the OTU1 MAG from IP, whereas B7 pathways were exclusively found in SAB MAGs from HS and not in IP or HP SAB MAGs. The menaquinone and tocopherol pathways were absent in all SAB MAGs and sponge transcriptomes and thus

are likely acquired from a different source by the holobionts. Analysis of vitamin/cofactor transporters revealed a counter-association with their biosynthetic pathways for B7 and, especially, for B1 (Fig. 4G), which suggests that auxotrophic SABs encode transporters to obtain compounds from the environment (presumably from other members of the microbiome).

Our analysis indicates that SABs can potentially complement the metabolism of host sponges with all essential amino acids and multiple vitamins.

## Taurine dissimilation pathways are present in the genomes of major SABs

To establish stable symbiotic relationships, the flow of metabolites might be reciprocal, i.e., symbiotic bacteria should gain from feeding on sponge-produced compounds. In search of such metabolic exchanges, we focused on taurine, a prevalent compound found in sponge tissues, though sponges have not yet been experimentally demonstrated to synthesize and release it (86, 87). Dissolved taurine is an important source of carbon and energy for seawater prokaryotic communities (88, 89), and, similarly, might be crucial for sponge symbionts (14).

Previously, taurine biosynthetic pathways were identified in the complete genome of the sponge *Amphimedon queenslandica* (90). Using KEGG, we revealed complete pathways for taurine biosynthesis from cysteine in the HP Tr2 and *Isodictya sp.* transcriptomes, supporting the potential host-derived origin of this compound (Fig. 5A). In contrast, all SAB MAGs lacked taurine biosynthetic pathways.

We found that MAGs of all dominant SABs (OTU4, OTU3, and OTU1) from all three sponge species encoded putative taurine ABC transporters (*tauABC*), suggesting the active import of taurine. Among these bacteria, OTU4 (HP) and OTU3 (HS) MAGs harbored genes encoding enzymes involved in taurine catabolism, converting it into sulfite and acetyl-CoA via sulfoacetaldehyde and acetyl-phosphate intermediates: taurine:pyruvate aminotransferase (*tpa,* the gene was expressed in the HP Tr1 (meta)transcriptome), sulfoacetaldehyde acetyltransferase (*xsc*), and phosphate acetyltransferase (*pta*). OTU1 MAG (IP) and phylogenetically related bacteria carried *xsc* and *pta* but lacked the *tpa* gene (Table S13; Fig. 5A). However, they encoded a ω-amino acid aminotransferase (EC 2.6.1.18) that converts taurine into sulfoacetaldehyde, thus substituting it for Tpa activity (91, 92) (Table S13; Fig. 5A). Additionally, we revealed metabolic genes from other pathways associated with taurine catabolism in the MAGs of the dominant SABs (Note S5). Notably, a complete pathway, including the *tauABC*, *tpa*, *xsc*, and *pta* genes, was also identified in 8 of 99 non-sponge-associated metagenomic bins retrieved from seawater and SAMs, all of which belonged to Alphaproteobacteria (Table S13).

The genomes of minor SABs lacked genes encoding taurine transporters. The OTU7 MAG (HS) carried the gene for taurine dioxygenase (*tauD*), which is responsible for taurine conversion to sulfite and aminoacetaldehyde (93), while the OTU23 MAG (HP) carried a complete pathway with the ω-amino acid aminotransferase, *xsc*, and *pta* genes present (Fig. 5B).

Our data indicate that taurine catabolism is widespread among taxonomically unrelated SABs, which suggests that this compound can potentially be utilized by SABs as a source of carbon, sulfur, and nitrogen.

Given that taurine may serve as an important nutrient source for SABs, we attempted to cultivate major SABs from HP and HS using various media, including poor and rich compositions supplemented with taurine (see the Supplemental methods). However, we did not obtain any colonies corresponding to OTU4 or OTU3 bacteria under any conditions, suggesting that taurine alone is not the limiting factor for their growth and that these bacterial residents are strongly associated with a host-specific environment.

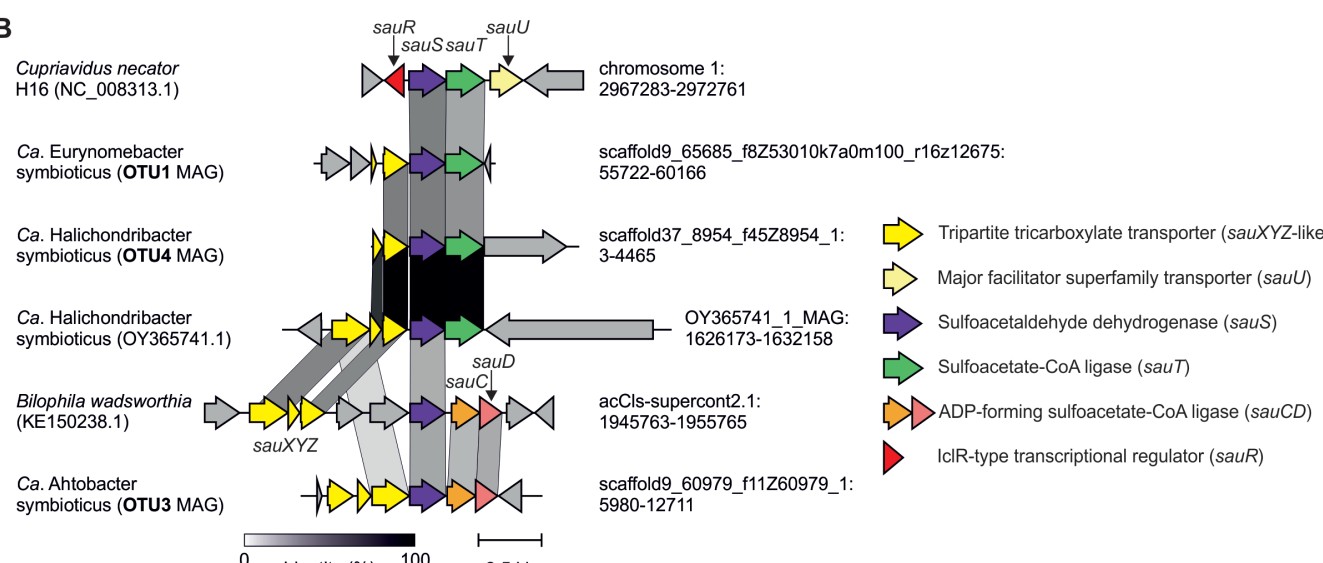

**FIG 5** Taurine and sulfoacetate metabolism pathways in sponges and SABs. (A) Metabolic map of taurine and sulfoacetate biosynthesis and degradation according to KEGG. The shading of the *tpa* gene in the OTU3 MAG indicates that this gene was found in several original OTU3 metagenomic bins but was absent in the final OTU3 MAG. (B) Comparison of characterized gene clusters involved in the degradation of sulfoacetate from *Bilophila wadsworthia* and *Cupriavidus necator* and putative gene clusters from major SABs. The chromosomal location of each cluster (chromosome ID: coordinates) is indicated to the right of the corresponding cluster.

## Major SABs encode putative gene clusters responsible for the uptake and degradation of the taurine derivative, sulfoacetate

Another compound linked to taurine metabolism is sulfoacetate, which can be produced by the deamination and oxidation of taurine by some marine bacteria during the assimilation of taurine nitrogen (94). This compound can also be imported and metabolized by other bacteria via the sulfoacetaldehyde pathway, allowing it to be used as a source of carbon and energy (95–97).

We noticed that all three dominant SABs encoded genes involved in the conversion of sulfoacetate to sulfoacetaldehyde (Fig. 5A). OTU4 (HP) and OTU1 (IP) MAGs carried gene clusters resembling the *sauST* cluster from *Cupriavidus necator* H16 and encoded sulfoacetaldehyde dehydrogenase (SauS) and sulfoacetate-CoA ligase (SauT) (96). The OTU3 MAG (HS) contained an alternative gene cluster with the *sauT* gene substituted with the *sauCD* (encoding the ADP-forming sulfoacetate-CoA ligase) genes, resembling a *sauSCD* cluster responsible for sulfoacetate processing in *Bilophila wadsworthia* (97). All the detected *sau* gene clusters were associated with genes encoding the tripartite tricarboxylate transporter (TTT), which is a putative sulfoacetate transporter, SauXYZ, previously identified in *B. wadsworthia*. The putative *sauST* gene clusters from the OTU1 and OTU4 MAGs therefore comprised a hybrid type associated with a TTT transporter instead of a single-gene-encoded SauU major facilitator superfamily transporter described in *C. necator* (96) (Fig. 5B).

Sulfoacetate metabolism genes were rare in non-sponge-associated metagenomic bins: *sauST* and *sauSCD* clusters were detected in 4/99 and 3/99 of such bins, respectively. Interestingly, four of these bins belong to the same orders as SABs from HP (Puniceispirillales or Rhodobacterales) (Table S13).

Although we did not detect the previously described *safD* pathway, responsible for sulfoacetate production in a marine bacterium (94), within the studied SABs or other metagenomic bins from SAMs, we hypothesize that alternative pathways exist within a holobiont and suggest that SABs may utilize the released sulfoacetate as an additional carbon and energy source.

## Genomes of SABs encode symbiosis-associated genes

Next, we focused on genes known to be associated with the symbiotic lifestyle. For example, bacterial ELPs are considered to be involved in symbiont-host interactions (15, 98) and enriched in sponge metagenomes and SAB genomes (16, 17, 19). To investigate this association, we predicted ELP domain-containing proteins in the reconstructed SAB MAGs and other medium-to-high-quality metagenomic bins from sponge and seawater metagenomes (Fig. S11).

We found that ELP domain frequencies, normalized by MAG length, were threefold higher in SAB MAGs in comparison to non-sponge-associated (control) metagenomic bins (2.2 vs 0.7 domains/Mb; *t*-test *P*-value 7.1e−9). Furthermore, PCA, performed on normalized ELP domain frequencies, separated SAB MAGs from one other and from the majority of control bins, indicating that SAB MAGs had different sets of enriched ELP domains (Fig. 6A). Indeed, tetratricopeptide repeat domains were abundant in the OTU1 (IP) and OTU9 (HS) MAGs, as their normalized frequencies were significantly higher compared to 99 control bins (*t*-test *P*-value 4.2e−2). Ankyrin domains (Ank) were substantially enriched in the OTU1 MAG (*t*-test *P*-value 1.7e−3). Fibronectin domains (Fn3) were associated with and significantly enriched in OTU14 MAG (HS) (*t*-test *P*-value 3.1e−4). Sel1-like repeats (Sel1) were highly enriched in the remaining (five out of seven) SAB MAGs (*t*-test *P*-value 8.0e−13) (Fig. 6B; Fig. S13A). Similar associations were observed for the frequencies of ELP-encoding genes (Fig. S13B and C). A substantial fraction of ELP-containing proteins (46/123 or 37%) from SAB MAGs were predicted to have signal peptides for translocation and secretion.

To further analyze adaptation of SAB MAGs to symbiotic lifestyle on a genomic level, we compared the normalized abundance of ELP domains encoded in SAB MAGs and genomes of taxonomically related bacteria (Table S13). We found that all SAB

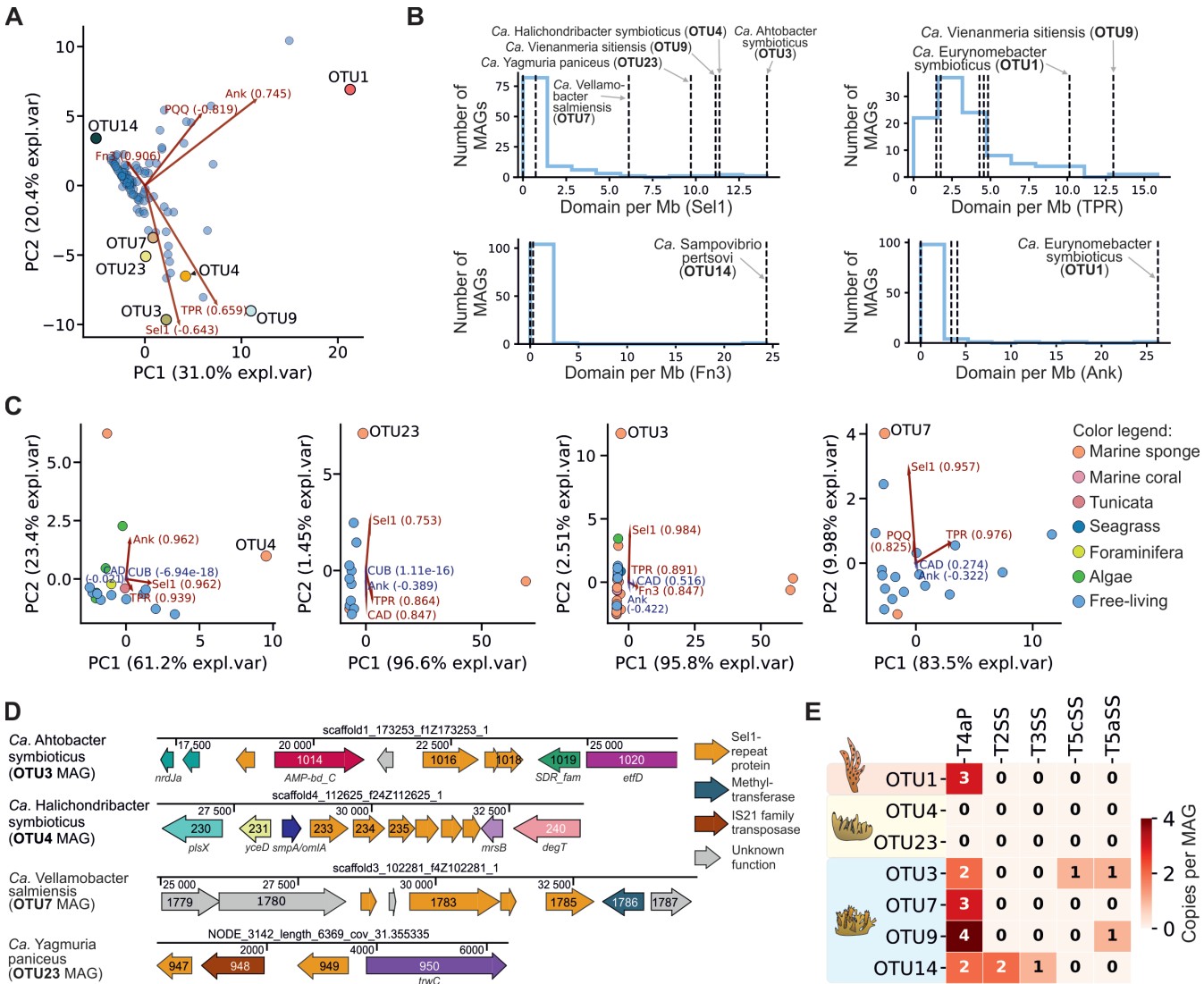

**FIG 6** Eukaryotic-like proteins and transport systems identified in SAB MAGs. (A) PCA biplot of normalized ELP domain frequencies in SAB MAGs (labeled dots) and control metagenomic bins (blue dots) reconstructed from SAM and seawater metagenomes. The number in parentheses indicates feature loading onto a principal component. (B) Distributions of normalized ELP domain frequencies in SAB MAGs and control metagenomic bins. Data are shown for ELPs enriched in SAB MAGs. The frequencies of ELPs in SAB MAGs are indicated with vertical dashed lines. (C) PCA biplots of normalized ELP domain frequencies in SAB MAGs (labeled dots) and taxonomically related bacteria. Isolation source of bacteria is color-coded. Data are shown for OTU4, OTU23, OTU3, and OTU7. (D) Gene clusters enriched with Sel1-repeat protein genes identified in SAB MAGs. (E) A heatmap representing the occurrence of secretion system and pili gene clusters in SAB MAGs.

MAGs, which had closely related free-living bacteria (OTU4, OTU23, OTU3, and OTU7), had genomes strongly enriched with Sel1 domains compared to their relatives (*t*-test *P*-value 0.01; Fig. 6C; Fig. S13D), while SAB MAGs from known symbiotic groups (OTU9—Porifericomitales, OTU14—Porisulfidales, and OTU1—AqS2/Tethybacterales) had other enriched ELPs (Fig. S13E and F). Notably, Sel1-encoding genes in SAB MAGs were frequently organized in non-conserved gene clusters located in variable genomic context (Fig. 6D).

We identified active expression of a *Ca*. H. symbioticus (OTU4 MAG from HP) gene encoding Sel1-containing protein with a predicted export signal peptide (Sec/SPI-type, score: 0.99) in the HP Tr1 (meta)transcriptome. Structural predictions using AlphaFold 3 and a similarity search using Dali revealed a high level of similarity to the exported Sel1-containing protein LpnE from *Legionella pneumophila*, which is required for the

invasion of eukaryotic cells (99). These data first confirm the accumulation of certain ELPs in SABs and, second, potentially indicate the role of Sel1 in mediating host-symbiont interactions and the association of *Ca*. H. symbioticus with HP cells.

Another group of pro-symbiotic factors includes secretion systems and other mechanisms involved in interactions with the extracellular matrix. Both bacterial symbionts and pathogens often encode such molecular tools, which allow them to interact with host cells (through adhesins, pili, curli, and fimbriae) or use secretion systems to export effector proteins that manipulate the host (100, 101). To study the repertoire of these systems in SAB MAGs, we utilized TXSScan. We observed that type 4 pili (T4aP) were encoded in all SAB MAGs, except HP-associated OTU4 and OTU23 (Fig. 6E). Interestingly, these two SABs did not encode any of the secretion system-related gene clusters, indicating an alternative mechanism of binding or interaction with host cells (e.g., mediated by exported ELPs). The absence of secretion system genes was unlikely due to the incompleteness of the MAGs, at least for *Ca*. H. symbioticus, because its complete genome did not contain any related gene clusters. Bacteria associated with HS encoded unique combinations of gene clusters, likely reflecting the specialization of these bacteria for different niches in the host organism.

Two paralogous genes from the OTU4 MAG (HP), both annotated as "Invasion protein B" (*ialB*), were found to be actively expressed in the HP Tr1 (meta)transcriptome. Homologs of these proteins were also identified in the OTU23 (HP) and OTU3 (HS) MAGs. Secretory signal sequences (Sec/SPI-type) were predicted in these proteins with high confidence (>0.97). *IalB* genes were previously detected in the genomes of *Pseudovibrio* bacteria isolated from different marine ecosystems, including the sponge *Polymastia penicillus* and other invertebrates (102, 103). *IalB* from the human pathogen *Bartonella bacilliformis* is a virulence factor (104) important for the invasion of erythrocytes. The expression of the *ialB* gene by sponge SABs may be important for host-microbe interactions, possibly mediating phagocytosis in sponge cells.

The persistence of identified SABs in sponge microbiomes over a 6 year period, combined with the analysis of their MAGs, revealed several key symbiotic features, including the enrichment of ELPs, suggesting that the studied SABs are sponge symbionts.

## Localization of SABs in sponge tissues

To orthogonally confirm the presence of identified SABs in sponge tissues and study their localization, we performed FISH microscopy using bacteria-specific oligonucleotides for sponge tissue sampled from different body sites—the osculum region, the middle, and the bottom sections of the sponge body. Cy3-labeled probes (red) were used for selected bacteria, and Cy5-labeled probes (green) were used for all bacteria. To control for probe specificity, we performed FISH with non-cognate sponge species and did not detect any signal under the same hybridization and imaging conditions (Fig. S14).

We found that SABs were present across all sampled body sites. The cells of the major HP SAB, OTU4, were small and rod-shaped, matching the morphology described previously for *Ca*. H. symbioticus (67). The bacteria were presumably accumulated within some host cells (bacteriocytes) and were also scattered in the mesohyl (Fig. 7A). We were unable to obtain a reliable image of SAB OTU23 (HP), likely due to its low abundance in the sponge (2% ± 1% based on the 16S rRNA gene amplicon sequencing data). HS tissues were densely populated with SABs. The dominant SAB of HS, OTU3, had rod-shaped cells (Fig. 7B). Other SABs, OTU7 and OTU9, also displayed a rod-shaped morphology, while OTU14 cells were crescent-shaped (Fig. 7D through F). The cells of OTUs 3, 9, and 14 were supposedly associated with bacteriocytes and located as individual cells intercellularly, while the OTU7 cells were primarily intercellular. A single SAB from IP, OTU1, dominated in the tissue samples in accordance with the 16S rRNA gene amplicon sequencing data. The OTU1 cells exhibited a spherical morphology and were likely clustered in intercellular microcolonies (Fig. 7C).

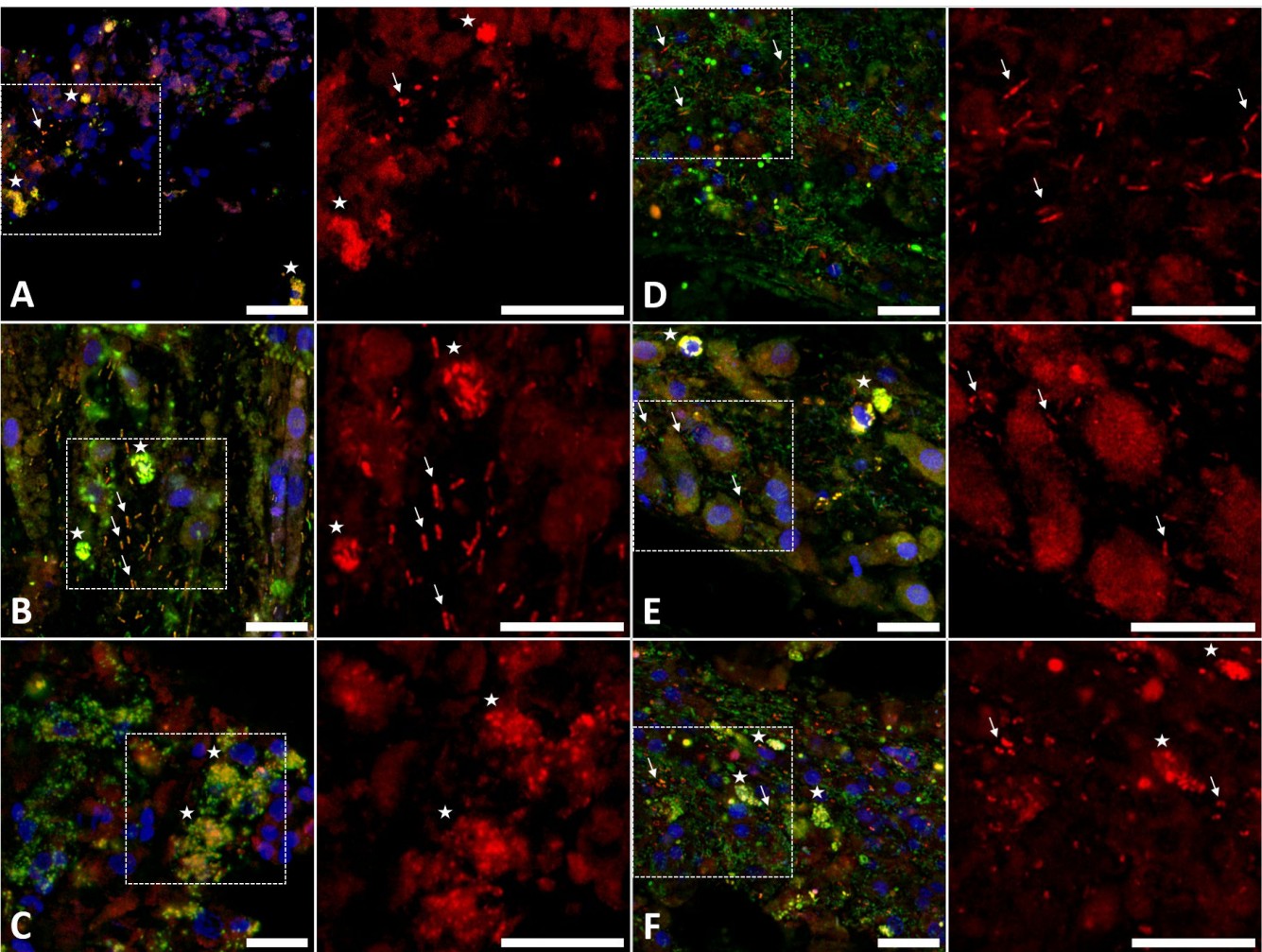

**FIG 7** FISH images of bacteria in tissue sections of the studied sponges. Bacteria were labeled with SABs-specific Cy3 (red) and universal probes with Cy5 (green). DNA was stained with Hoechst 33342 (blue). (A) Section of HP hybridized with a probe specific for OTU4, (B) section of HS hybridized with a probe specific for OTU3, (C) section of IP hybridized with a probe specific for OTU1, (D) section of HS hybridized with a probe specific for OTU7, (E) section of HS hybridized with a probe specific for OTU9, (F) section of HS hybridized with a probe specific for OTU14. In all, a merged image of the three color channels is shown on the left; an enlarged field, indicated by a white rectangle in the merged image, is shown on the right in the red (Cy3) channel only. Arrows and asterisks indicate individual cells and their clusters, respectively. Scale bars: 10 µm.

Taken together, we detected all identified SABs (except OTU23) in different body sites of the sponges and found that they exhibited distinct patterns of tissue localization.

## DISCUSSION

In the present study, we investigated the microbiomes of three species of Arctic LMA sponges. We identified a spectrum of increasingly complex associated bacterial communities consisting of one (IP), two (HP), and four (HS) species of symbiotic bacteria. We repeatedly retrieved the identified symbionts from sponges in the WSBS collection site (the White Sea) over a 6 year period, indicating their stable and species-specific associations. The identified bacteria belong to one novel family, three novel genera, and six novel species of SABs from the Alpha- and Gammaproteobacteria phyla.

Although the SAMs were consistently dominated by symbiotic OTUs throughout the observation period, some samples collected in 2018 were compromised by an increased abundance of Alteromonadales and Vibrionales OTUs, which had been scarce in earlier (2016) and later (2022) collections. These changes may be linked to a mass mortality

of sponges, detected in the summer of 2018 near WSBS MSU, which was presumably associated with abnormally high water temperatures (28, 105). While cold-water sponges demonstrated some capacity to sustain stable microbiomes at elevated temperatures (65), our data potentially indicate that acute thermal stress may affect SAM composition in a manner similar to tissue injuries or antibiotics (69, 70). However, a small sample size and the lack of replication prevent establishing a strong functional connection between environmental stress and changes in SAMs, as well as underlying mechanisms, which are the topics for future studies.

The detection of sponge-specific symbiotic OTUs in seawater and other sponge species in 2018 at the WSBS collection site may indicate the release of viable bacteria from decaying sponges. However, this hypothesis is based on a limited number of samples and requires further experimental validation. If confirmed, it could imply broader consequences of heat-induced sponge stress, including enhanced exchange of symbiotic bacteria between different sponge hosts and the surrounding environment.

The identified sponge symbionts are species-specific. Indeed, we observed limited exchange between sympatric sponge species without establishing new symbiotic interactions and symbiont proliferation in a new host. This finding suggests the existence of mechanism(s) allowing discrimination between self and non-self symbiotic bacteria even for closely related species of sponges (like HP and HS). Discrimination and exclusion may be mediated by (i) a resident symbiotic community (e.g., an active defense against invaders); (ii) specific conditions in a host microenvironment (e.g., availability of specific metabolites or presence of viruses); (iii) an interaction between a symbiont and a sponge (e.g., specific recognition by the immune system or specific suppression by the immune system), or a combination of these factors. Natural products with antibacterial activities encoded by biosynthetic gene clusters (NP BGCs) are widespread bacterial tools in competition for resources (106, 107). Previously, diverse NP BGCs were reported in sponge symbiont MAGs, reaching 3.5 NP BGCs per MAG (108, 109). The symbiotic MAGs recovered in the current work were particularly poor in NP BGCs (<1 BGC/MAG), which makes the first hypothesis unlikely. However, this does not rule out the presence of yet unidentified NP BGCs that remain undetected by standard algorithms such as antiSMASH, nor does it exclude the possibility of alternative defense strategies. Although the exact conditions within the tissues of the different studied sponges are not known, transcriptomic, genomic, and ecological data indicate that they might be similar. Indeed, symbiotic MAGs recovered from different sponges showed similar metabolic patterns. For the third scenario, the mechanisms of interactions between bacterial symbionts and sponges have not yet been studied. It was proposed that sponges may recognize specific bacteria via microbial-associated molecular patterns using their pattern recognition receptors, such as nucleotide oligomerization domain-like receptors (58, 110, 111). On the other hand, symbiont-encoded ELPs were considered potential modulators of symbiont-host interactions (15, 98). We found that symbiotic MAGs were significantly enriched in ELPs, and a substantial fraction of them (37%) carried a secretion signal. We hypothesize that such exported ELPs (and Sel1-containing proteins, specifically) may be modulators of the host immunity system, altering the response to general microbial-associated molecular patterns and licensing the presence of symbionts in sponge tissues or cells. Given the similar ecology of the three sympatric sponge species, which harbor relatively simple yet distinct SAMs, we propose them as a model system for further functional studies of sponge-symbiont interactions and their specificity.

Earlier works on the chemotaxonomic analysis of sponges used taurine content as a taxonomical marker (112, 113). Particularly, both Halichondrida (including HS and HP) and Poecilosclerida (including IP) were characterized by high taurine contents of >18% (86). It has never been experimentally confirmed that sponges are responsible for taurine synthesis; however, the transcriptomics and genomics of sponges have revealed the presence of all the genes necessary for this process. Several studies have identified genes responsible for taurine import and processing in the genomes of sponge symbiotic bacteria (14, 114–116). Here, we further expand the understanding of the association

mSystems

between sponge symbiosis and taurine metabolism capabilities by demonstrating the presence of complete pathways for taurine import and dissimilation in three taxonomically distinct species of dominant sponge symbionts. However, the metabolism of taurine is widespread in free-living marine bacteria (88, 89, 117). Particularly, 8% of the bins retrieved from seawater metagenomes in the current study carried complete taurine metabolism pathways. Therefore, taurine catabolism capabilities are likely not an exclusive feature of symbiosis but may be critical for symbiotic species.

In addition to taurine dissimilation pathways, we detected two types of gene clusters presumably responsible for sulfoacetate uptake and dissimilation in the genomes of three major symbionts. It is currently unclear how sulfoacetate is generated in sponge holobionts. It can potentially be excreted by sponges or produced by other members of the microbiome as a byproduct of taurine deamination (94, 118). In the latter case, cross-feeding between bacterial species is expected. Additionally, sulfoacetate can be acquired from seawater where it is presumably synthesized alongside other organosulfonates by bacterioplankton (119). Further research is needed to determine how widespread sulfoacetate metabolism is among different taxonomic groups of sponge symbionts. The sulfoacetate catabolic pathway may be an indicative, though not exclusive, feature of sponge symbionts, similar to taurine metabolism.

Analysis of sponge microbiomes featuring different numbers of bacterial symbionts allowed us to identify functional commonalities between microbiomes and specialization within them at the species level. As such, all symbiotic microbiomes are likely to use carbohydrates as a primary source of energy via oxidative pathways. All the symbiotic genomes were enriched with ELPs; however, they had different types of ELPs, even for symbionts of phylogenetically close HP and HS. Furthermore, different complete biosynthetic pathways of amino acids and vitamins/cofactors were encoded in different symbionts, especially in the case of HS symbionts (most prominently, for polar amino acids and B vitamins). Finally, complete taurine and sulfoacetate catabolic pathways were found exclusively in the genomes of dominant symbionts but not in minor symbionts. A specific example of specialization is the sulfur metabolism of HS symbionts. While OTU3, OTU9, and OTU14 encoded various sulfur oxidation pathways, OTU7 encoded the assimilatory sulfur reduction pathway and may benefit from sulfate produced by other symbionts. Taken together, these observations are concordant with a microbiome model in which symbionts occupy minimally overlapping niches and cooperate rather than compete for resources (120, 121). Further studies are needed to discover the roles of ELPs in the possible specialization of symbionts and to identify potential cross-feeding between the species. Overall, our data suggest that bacterial symbionts may utilize taurine and/or sulfoacetate synthesized by sponges as carbon and energy source while simultaneously providing them with amino acids and vitamins, thereby complementing host metabolism and contributing to the holobiont.

## ACKNOWLEDGMENTS

The authors would like to acknowledge the personnel of the MSU White Sea Biological Station, particularly Dr. Alexander Tzetlin and Dr. Tatyana Neretina for the opportunity to collect and process samples at WSBS, the WSBS scuba diving team for sample collection, and Boris Osadchenko for the identification of sponge species. We would like to thank the staff of the MRC MSU for their assistance in collecting samples from the Barents Sea. We would like to thank Dr. Svetlana Dubiley (TBI, France), Dr. Dmitrii Travin (UIC, USA), and Anika Gupta (UW, USA) for fruitful discussions and critical reading of the manuscript. FISH microscopy was performed at the Core Centrum of the Institute of Developmental Biology RAS.

Y.L. and A.F. were supported by the IDB RAS Government basic research program in 2024 (no. 0088-2024-0009). Sequencing in the Skoltech Genomics Core Facility was supported by an internal organization grant.

## AUTHOR AFFILIATIONS

[1]The Center for Molecular and Cellular Biology, Moscow, Russia

[2]Institute of Gene Biology Russian Academy of Sciences, Moscow, Russia

[3]Koltzov Institute of Developmental Biology, Russian Academy of Sciences, Moscow, Russia

[4]Principal Engineering School, ITMO University, Saint Petersburg, Russia

[5]Lomonosov Moscow State University, Moscow, Russia

[6]A.A. Kharkevich Institute for Information Transmission Problems, Russian Academy of Sciences, Moscow, Russia

## AUTHOR ORCIDs

Anastasiia Rusanova http://orcid.org/0000-0002-1674-5635

Dmitry Sutormin http://orcid.org/0000-0001-6834-4246

## FUNDING

| Funder | Grant(s) | Author(s) |
| --- | --- | --- |
| IDB RAS Government basic research program | № 0088-2024-0009 | Alexander Finoshin |
| IDB RAS Government basic research program | № 0088-2024-0009 | Yulia Lyupina |

## AUTHOR CONTRIBUTIONS

Anastasiia Rusanova, Conceptualization, Data curation, Formal analysis, Investigation, Methodology, Visualization, Writing – original draft, Writing – review and editing | Viktor Mamontov, Formal analysis, Investigation, Software | Maxim Ri, Formal analysis, Software | Dmitry Meleshko, Formal analysis, Software | Anna Trofimova, Investigation, Resources | Victor Fedorchuk, Data curation, Investigation | Margarita Ezhova, Data curation, Formal analysis, Investigation | Alexander Finoshin, Visualization | Yulia Lyupina, Visualization | Artem Isaev, Funding acquisition, Writing – review and editing | Dmitry Sutormin, Conceptualization, Data curation, Formal analysis, Funding acquisition, Investigation, Methodology, Project administration, Resources, Software, Supervision, Visualization, Writing – original draft, Writing – review and editing

## DATA AVAILABILITY

The raw sequencing data are publicly available in the NCBI Sequence Read Archive (SRA) under the Project Accession number PRJNA1211653. MAGs can be accessed via the BioSample accession numbers SAMN46784898–SAMN46784904. The scripts, used for data visualization and source data for metabolism and ELP analyses, are available on GitHub (https://github.com/sutormin94/Sponge_metagenomes_2024). The CORITES algorithm is available on GitHub (https://github.com/sutormin94/CORITES).

## ADDITIONAL FILES

The following material is available online.

### Supplemental Material

**Supplemental Material (mSystems01147-25-s0001.pdf).** Supplemental methods, notes, and figures.

**Supplemental Tables (mSystems01147-25-s0002.xlsx).** Tables S1-S13.

### Open Peer Review

**PEER REVIEW HISTORY (review-history.pdf).** An accounting of the reviewer comments and feedback.

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
