## [Reviewer comments · mSystems]

Taxonomically different symbiotic communities of sympatric Arctic sponge species show functional similarity with specialization at species level

Anastasiia Rusanova, Viktor Mamontov, Maxim Ri, Dmitry Meleshko, Anna Trofimova, Victor Fedorchuk, Margarita Ezhova, Alexander Finoshin, Yulia Lyupina, Artem Isaev, and Dmitry Sutormin

Corresponding Author(s): Dmitry Sutormin, The Center for Molecular and Cellular Biology

Review Timeline:

Submission Date:	August 4, 2025
Editorial Decision:	September 7, 2025
Revision Received:	September 19, 2025
Accepted:	September 23, 2025

Editor: Ariane Peralta

Reviewer(s): The reviewers have opted to remain anonymous.

Transaction Report:

DOI: <https://doi.org/10.1128/msystems.01147-25>

Re: mSystems01147-25 (**Taxonomically different symbiotic communities of sympatric Arctic sponge species show functional similarity with specialization at species level**)

Dear Dr. Dmitry Sutormin:

Revision Guidelines

Sincerely,
Ariane Peralta
Editor
mSystems

Reviewer #1 (Comments for the Author):

I appreciate Authors' consideration of the comments and the provided detailed replies, including additional analysis. Details of methods that were missing are now provided. The revised structure of the results has greatly improved the manuscript's flow and readability.

From the revised version, I have only one pending comment:

The Authors acknowledge the need of more replicates and additional sampling to validate their conclusions regarding temporal

stability of sponge-associated microbiota. Although in the revised discussion they explicitly acknowledge this limitation and the need for further studies (line 833) and the need for further studies (line 812-814; 828-830; 833-834), the Authors keep mainly the original (lengthy) discussion and even add some additional hypothesis (line 746-751). Considering the low replication, this long discussion is highly speculative and, importantly, it distracts the reader from the main results of this study. Therefore, I strongly advice to reduce this paragraph (lines 801 to 836) to 1) report the repeatedly retrieval of certain OTUs, 2) report the differences in abundance for certain groups and 3) suggest in 1-2 sentences that these changes may be due to mass mortality events in the summer of 2018, 4) acknowledge that the lack of replication prevents establishing a connection between the microbial changes and environment and thus, it is the topic for future studies. That makes a short paragraph of 6-7 sentences. The statement on "dramatic shifts in the compositions of sponge microbiomes coinciding with abnormally elevated water temperatures during the 2018 season" (lines 59-61) should be removed from the abstract.

Minor comments:

L452: "no nitrogen metabolism pathways were detected in the MAGs". Please specify which pathways you mean (e.g., autotrophic metabolism?)

Reviewer #2 (Comments for the Author):

I acknowledge and appreciate the effort made in addressing all the concerns raised in the review. The authors have provided satisfactory responses and have implemented the necessary revisions to improve the clarity, methodological rigor, and overall quality of the manuscript. I have no further questions or requests for modification. I wish to congratulate the authors for the effort and the nice paper presented.

RESPONSES TO THE REVIEWERS' COMMENTS

Reviewer #1

The Reviewer “appreciated Authors' consideration of the comments and the provided detailed replies, including additional analysis.” The Reviewer stated that “details of methods that were missing are now provided” and “the revised structure of the results has greatly improved the manuscript's flow and readability.”

The Reviewer “has only one pending comment”:

1. The Authors acknowledge the need of more replicates and additional sampling to validate their conclusions regarding temporal stability of sponge-associated microbiota. Although in the revised discussion they explicitly acknowledge this limitation and the need for further studies (line 833) and the need of further studies (line 812-814; 828-830; 833-834), the Authors keep mainly the original (lengthy) discussion and even add some additional hypothesis (line 746-751). Considering the low replication, this long discussion is highly speculative and, importantly, it distracts the reader from the main results of this study. Therefore, I strongly advice to reduce this paragraph (lines 801 to 836) to 1) report the repeatedly retrieval of certain OTUs, 2) report the differences in abundance for certain groups and 3) suggest in 1-2 sentences that these changes may be due to mass mortality events in the summer of 2018, 4) acknowledge that the lack of replication prevents establishing a connection between the microbial changes and environment and thus, it is the topic for future studies. That makes a short paragraph of 6-7 sentences. The statement on "dramatic shifts in the compositions of sponge microbiomes coinciding with abnormally elevated water temperatures during the 2018 season" (lines 59-61) should be removed from the abstract.

We acknowledge the Reviewer's efforts to improve our manuscript. The section was shortened and rephrased following the Reviewer's suggestions:

“We repeatedly retrieved the identified symbionts from sponges in the WSBS collection site (the White Sea) over a 6-year period, indicating their stable and species-specific associations. The identified bacteria belong to one novel family, three novel genera, and six novel species of SABs from the Alpha- and Gammaproteobacteria phyla.

Although the SAMs were consistently dominated by symbiotic OTUs throughout the observation period, some samples collected in 2018 were compromised by an increased abundance of Alteromonadales and Vibrionales OTUs, which had been scarce in earlier (2016) and later (2022) collections. These changes may be linked to a mass mortality of sponges, detected in the summer of 2018 near WSBS MSU, which was presumably associated with abnormally high water temperatures [28, 105]. While cold-water sponges demonstrated some capacity to sustain stable microbiomes at elevated temperatures [65], our data potentially indicate that acute thermal stress may affect SAM composition in a manner similar to tissue injuries or antibiotics [69, 70]. However, a small sample size and the lack of replication prevent establishing a strong functional connection between environmental stress

and changes in SAMs, as well as underlying mechanisms, which are the topics for future studies.”

The statement on "dramatic shifts in the compositions of sponge microbiomes coinciding with abnormally elevated water temperatures during the 2018 season" was removed from the abstract.

Minor comments:

1. L452: "no nitrogen metabolism pathways were detected in the MAGs". Please specify which pathways you mean (e.g., autotrophic metabolism?)

A list of analyzed pathways was added to lines 441-442 as follows:

“No nitrogen metabolism pathways (such as nitrogen fixation, nitrification, denitrification, nitrate reduction, or anammox) were detected in the MAGs [73].”

Reviewer #2

The Reviewer “acknowledged and appreciated the effort made in addressing all the concerns raised in the review.” The Reviewer stated that “The authors have provided satisfactory responses and have implemented the necessary revisions to improve the clarity, methodological rigor, and overall quality of the manuscript.” The Reviewer “has no further questions or requests for modification” and “wish to congratulate the authors for the effort and the nice paper presented.”

Re: mSystems01147-25R1 (**Taxonomically different symbiotic communities of sympatric Arctic sponge species show functional similarity with specialization at species level**)

Dear Dr. Dmitry Sutormin:

Your manuscript has been accepted, and I am forwarding it to the ASM production staff for publication. Your paper will first be checked to make sure all elements meet the technical requirements. ASM staff will contact you if anything needs to be revised before copyediting and production can begin. Otherwise, you will be notified when your proofs are ready to be viewed.

Sincerely,
Ariane Peralta
Editor
mSystems